# Electric-field-enhanced second-harmonic domain contrast and nonreciprocity in a van der Waals antiferromagnet

Ziqian Wang [1] ✉, Meng Wang [1,2], Jannis Lehmann [1,3], Yuki Shiomi[4], Taka-hisa Arima[1,5], Naoto Nagaosa [1], Yoshinori Tokura [1,6,7] & Naoki Ogawa[1]

Imaging antiferromagnetic 180° domains with actively controlled visibility is vital for both fundamental science and sophisticated applications. While optical second-harmonic generation (SHG) is a well-known technique for distinguishing such domains in non-centrosymmetric antiferromagnets, a general material-based strategy to control domain contrast remains elusive. Using van der Waals antiferromagnet $MnPS_3$ as a proof of concept, we demonstrate the tuning of nonreciprocity-induced domain contrast in SHG through applying an in-plane electric field that transforms the magnetic point group to its unitary subgroup. The interference among intrinsic electric-dipole, magnetic-dipole, and field-induced electric-dipole transitions, each carrying distinct characters under space-inversion ($\mathcal{P}$) and time-reversal ($\mathcal{T}$) operations, enables large tuning of domain contrast and nonreciprocity in a broad spectral range. This strategy, generically applicable to systems characterized by $\mathcal{PT}$-symmetric magnetic groups with a polar unitary subgroup, offers a path to fast electrical modulation of nonlinear nonreciprocal photonic behaviors using antiferromagnets.

Antiferromagnets (AFMs) offer vast opportunities for advanced spintronics owing to their fast spin dynamics, stability against external magnetic fields, and potential for device miniaturization[1]. Recent discoveries and proposals of unique topological properties of AFMs, including anomalous charge and magnon transport, domain-wall conduction, etc., hold promise for novel device principles[2–4]. Given the inherent domain dependence of these effects, precise and efficient visualization of the antiferromagnetic domain structure is crucial. However, only a limited number of techniques are applicable to differentiate antiferromagnetic 180° domains, which differ solely by spin-flips[5]. Optical second-harmonic generation (SHG) is an important example providing such domain contrast for non-centrosymmetric AFMs, originating from optical nonreciprocity, i.e., the different optical response in materials when light propagation direction is reversed[6–8]. Nevertheless, effective and flexible enhancement of such contrast remains mostly unachieved, and a general strategy for controlling the domain-contrastive SHG yield and nonreciprocity on a material basis is yet to be established.

The recent emergence of two-dimensional (2D) van der Waals (vdW) AFMs with intriguing electron-correlation physics sets the stage for novel antiferromagnetic spintronics at atomic thicknesses[9–18]. $MnPS_3$, with magnetic order that breaks space-inversion symmetry, is predicted to exhibit diverse domain-dependent nonreciprocal effects[19,20], with the domains being identifiable through SHG[21,22]. In this work, with $MnPS_3$ as a proof of concept, we showcase the electrical modulation of AFM domain contrast (reaching approximately ± 90%) and associated nonreciprocity by tuning the optical-transition interference, resulting from the

[1]RIKEN Center for Emergent Matter Science (CEMS), Wako, Japan. [2]School of Integrated Circuits and Electronics, MIIT Key Laboratory for Low-Dimensional Quantum Structure and Devices, Beijing Institute of Technology, Beijing, China. [3]Department of Physics, ETH Zurich, Zurich, Switzerland. [4]Department of Basic Science, University of Tokyo, Tokyo, Japan. [5]Department of Advanced Materials Science, University of Tokyo, Kashiwa, Japan. [6]Department of Applied Physics, University of Tokyo, Tokyo, Japan. [7]Tokyo College, University of Tokyo, Tokyo, Japan. ✉e-mail: ziqian.wang@riken.jp

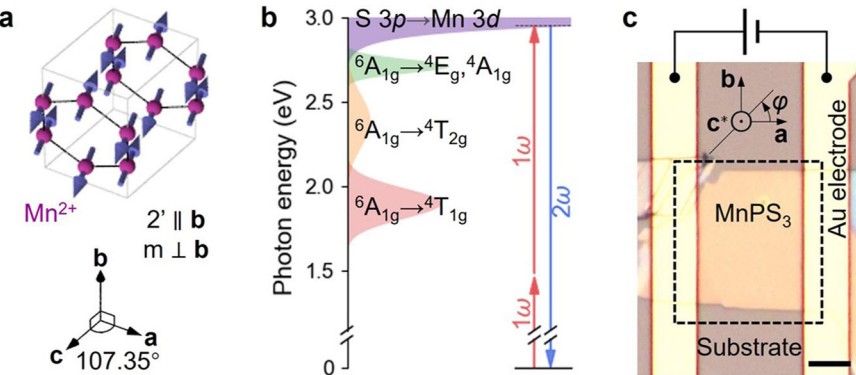

**Fig. 1 | Material and experimental setup. a** Schematic magnetic structure of MnPS$_3$ with blue arrows indicating spin on a Mn$^{2+}$ ion. **b** Optical energy diagram with photon energy $2\omega$ in resonance with charge-transfer excitation. **c** Microscopy image of our device configuration. Scale bar: 10 μm. An electric field is applied along the **a**-axis of a MnPS$_3$ flake on a sapphire substrate. The angle $\varphi$ defines the polarization direction (P) of the fundamental light wave. The dashed rectangular region is imaged by SHG, see Fig. 2.

transformation of the magnetic point group to its unitary subgroup. The distinct space-inversion ($\mathcal{P}$) and time-reversal ($\mathcal{T}$) odd-even properties of intrinsic electric-dipole (ED), magnetic-dipole (MD) and electric-field-induced electric-dipole (ΔED) transitions involved in SHG processes allow them to interfere differently for opposite domains, enabling electrical control. Our symmetry-based approach does not rely on addressing narrow optical transitions and can be applied over a broad spectral range to many $\mathcal{PT}$-symmetric systems with a magnetic point group that features a polar unitary subgroup. Our results give a renewed perspective on domain-contrastive nonreciprocal nonlinear photon responses via intra-material interference tuning and enable the design of novel compact 2D-material-based nonreciprocal photonic devices with high-speed controllability.

## Results

### Experimental setup

MnPS$_3$ has a near-honeycomb lattice within each layer ($ab$-plane) and a monoclinic stacking of the layers along the $c$-axis, leading to C2/m space group symmetry (Fig. 1a)[23]. Below the Néel temperature ($T_N \sim 78$ K for a bulk crystal), the antiferromagnetic ordering of Mn$^{2+}$ ions ($S = 5/2$) breaks the inversion symmetry, resulting in magnetic space group C2′/m[23]. The S 3$p$ to Mn 3$d$ charge-transfer transition dominates its bandgap, and multiple Mn$^{2+}$ $d$-$d$ intra-ionic transitions exist in the bandgap (Fig. 1b)[24,25]. To effectively demonstrate the domain-specific electrical control of SHG, we use an incident light wavelength of 840 nm (photon energy of $\hbar\omega = 1.48$ eV), such that the second harmonic (SH) is in resonance with $p$-$d$ charge-transfer transition and the fundamental approaches to the energy of the lowest $d$-$d$ transition (Fig. 1b). In this setting, both the ED process, $P_i(2\omega) \propto \chi_{ijk}^e E_j(\omega) E_k(\omega)$, and the MD process, $P_i(2\omega) \propto \chi_{ijk}^m E_j(\omega) H_k(\omega)$, contribute to the SHG response. Here, $\chi_{ijk}^e$ and $\chi_{ijk}^m$ denote the second-order nonlinear optical susceptibility tensors that relate the SH electric polarization to the fundamental electric, and electric and magnetic fields, respectively. (We collectively refer to the MD and electric-quadrupole (EQ) contributions to SHG as MD throughout this paper due to their equivalent symmetries.)

Our device, consisting of a MnPS$_3$ flake exfoliated onto a sapphire substrate, is shown in Fig. 1c. Two gold electrodes were deposited on the sample to enable the application of electric fields along the **a**-axis while preserving the mirror in the $ac$-plane. This configuration allows effective interference between the ΔED and the intrinsic ED and MD processes, as shown later. The sample was illuminated with fundamental light at normal incidence with the polarization P (defined by the angle $\varphi$ relative to the $a$-axis), and

imaged by the transmitted SH light after an analyzer A at an angle parallel or perpendicular to P.

### Electric-field-modulated SHG domain imaging

In the middle panels of Fig. 2a, b, we present exemplary SHG images acquired with P ∥ A, $\varphi = 0°$ and P ⊥ A, $\varphi = 90°$ configurations under electric fields of either sign at a sample temperature of 10 K. The electric field transforms the magnetic space group from C2′/m to Cm. Regions A and B, corresponding to higher and lower SH intensities in the absence of an electric field, are identified as antiferromagnetic 180° domains, as confirmed by their emergence below $T_N$ with varying shapes and locations after each cooling cycle (see Supplementary Fig. 2 for further exemplification). The SHG rotational anisotropy (RA) patterns for the two domains at the corresponding fields are displayed next to the images of the domains. At all applied electric fields, the patterns preserve the mirror symmetry along the horizontally aligned $a$-axis, as expected from the magnetic space group Cm under electric fields. Remarkably, the electric field stretches/compresses the patterns at positive/negative fields for Domain A, while the trend reverses for Domain B, as indicated by the gray arrows in Fig. 2a, b. Owing to the opposite responses of Domains A and B to the electric field, we can control the domain contrast effectively.

### Interference among multipolar SHG sources

These observations exemplify a general strategy for the electrical control of SHG domain contrast and nonreciprocity, as explained in Fig. 3. This control is based on the interference of intrinsic MD SHG with axial i-type $\chi_{ijk}^m$, intrinsic ED SHG with polar c-type $\chi_{ijk}^e$, and field-induced ED component (ΔED) with polar i-type $\Delta\chi_{ijk}^e (= \chi_{ijkl}^e E_l)$ (see Table 1 for details). Here, c or i signifies the sign-change or invariance of different multipole SHG contributions (represented by $\chi$'s) under time reversal, while polar or axial refers to this property under space-inversion. Both time-reversal and space-inversion lead to domain switching. With [Domain, electric field; incident wavevector] denoting the SHG conditions, domain contrast arises from the intensity difference between $[A, +E; +\mathbf{k}_{1\omega}]$ and $[B, +E; +\mathbf{k}_{1\omega}]$ (right and left panels of Fig. 3a), whose interferences are presented by the sum of $\chi$'s schematically shown on the complex plane (Fig. 3c, d). Specifically, in the $|ED| > |MD|$ regime, at $E = 0$, the sign-change of ED contribution under time-reversal (domain switching) results in $\chi$-sums in opposite quadrants (Point 2 in Fig. 3c, d), exhibiting different SH amplitudes (distances from the origin to Point 2) for Domains A and B. Applying $+E$ shifts the $\chi$-sums from Point 2 to Point 3, resulting in enhanced and suppressed SH intensities (amplitudes squared) for Domains A and B (Fig. 3c, d), thus enabling the electrical modulation of domain contrast.

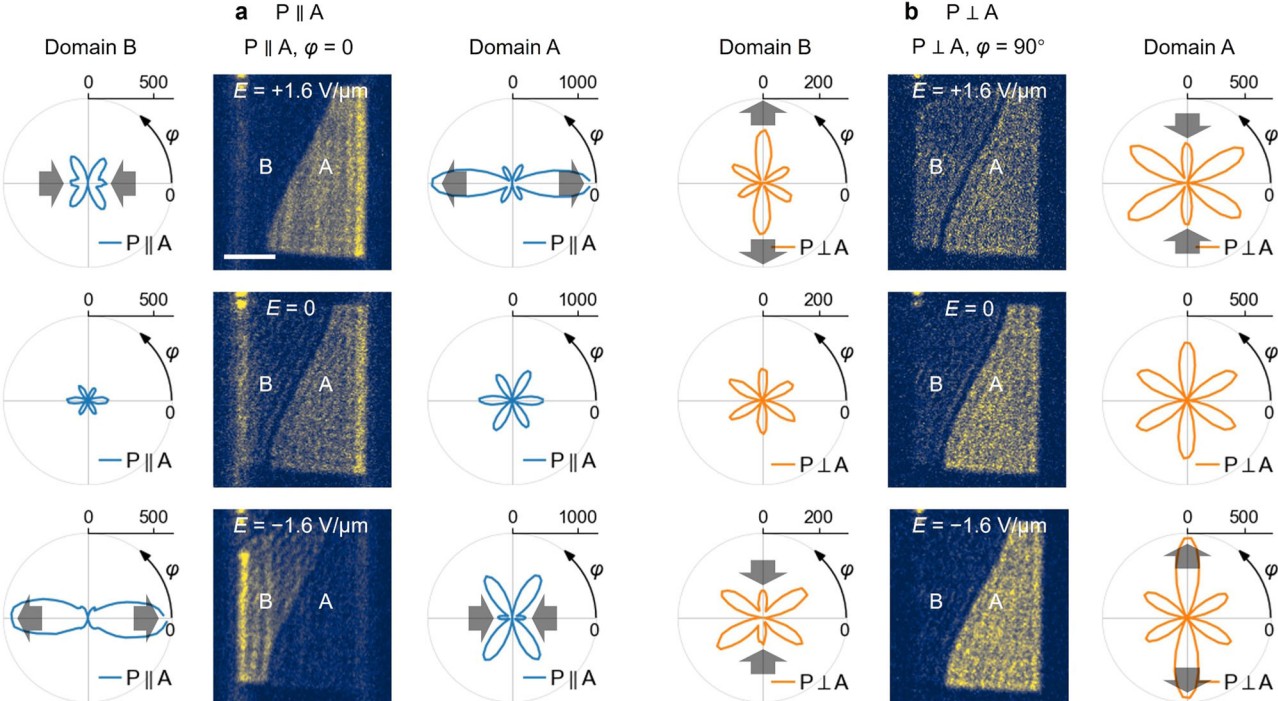

**Fig. 2 | Electrical control of SHG domain contrast. a, b** Co- (P ∥ A) and cross- (P ⊥ A) polarization configurations (P: polarizer, and A: analyzer). Middle columns in (**a, b**) display SHG images with controlled domain contrast acquired at 10 K under the change of applied electric fields. Scalebar: 10 μm (shared by all images). Corresponding SHG rotation anisotropy (SHG-RA) patterns for Domains A and B are shown on the right and left sides of the images, respectively. Opposite electric fields yield contrary effects on the same domain, while the same field results in opposing effects on different domains, as indicated by the gray arrows, for both polarization configurations.

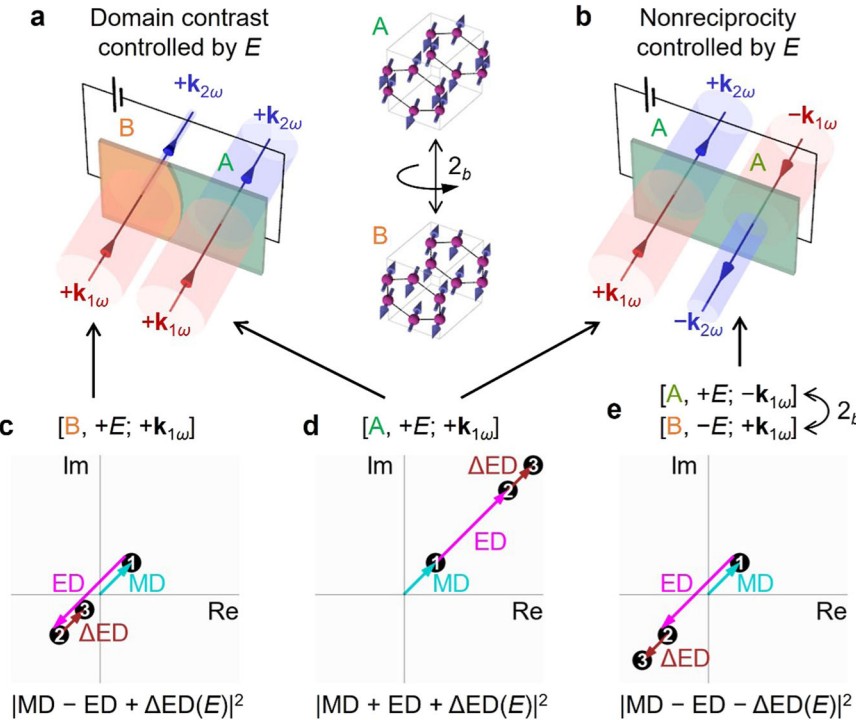

**Fig. 3 | Principle of electrical control of SHG domain contrast and non-reciprocity via the interference of optical transitions. a** Intensity contrast between Domains B (left) and A (right) due to the interference of complex SHG sources shown in (**c**) and (**d**), respectively. **b** Unequal efficiency for the forward (left) and reversed (right) light paths arising from the interferences shown in (**d**) and (**e**), respectively. **c**–**e** Conceptual illustration of MD, ED, and field-dependent ΔED transitions interfering on the complex plane. Endpoints 1 to 3 provide guidance for vector summation in the interference process. The distance from the origin to Point 3 gives the SH amplitude, whose square leads to SH intensity. Point 3 coincides with Point 2 at $E = 0$. The two labels in (**e**) denote the same physical process viewed from front and back. Domains A and B interchange upon a $2_b$-operation which is equivalent to their observation from opposite sides (front/back), as shown between (**a**) and (**b**).

Moreover, nonreciprocity stems from the distinction between $[A, +E; +\mathbf{k}_{1\omega}]$ and $[A, +E; -\mathbf{k}_{1\omega}]$ (left and right panels of Fig. 3b), with the latter being equivalent to $[B, -E; +\mathbf{k}_{1\omega}]$ after 180° rotation of the entire system around the $b$-axis. The sign reversal of ED and ΔED under space-inversion leads to different interference scenarios for $[A, +E; +\mathbf{k}_{1\omega}]$ and $[B, -E; +\mathbf{k}_{1\omega}]$ (Fig. 3d, e), where the field-dependent shift of Point 3 enables the electrical control of nonreciprocity as well. Here, photon polarization is omitted since $\varphi$ and $-\varphi$ conditions are always equivalent due to the mirror symmetry. Note that the presentations in Fig. 3c–e are conceptual. In practice, the three vectorial contributions may not align in a straight line on the complex plane due to specific resonance conditions. Also, multiple sets of such interference could coexist, depending on the number of active $ijk$ indices.

For a quantitative understanding and comparison with our scheme, we performed fitting analyses on the SHG-RA response according to the symmetry-adapted SH intensity formula as given in Eq. (1) (see Supplementary Note 1 for details of its derivation).

$$I_{\mathrm{SH}} \propto \begin{cases} \left| \left( \chi_a \cos^2\varphi + \chi_b \sin^2\varphi \right) \cos\varphi + \chi_c \sin 2\varphi \sin\varphi \right|^2 & P \parallel A \\ \left| \left( \chi_a \cos^2\varphi + \chi_b \sin^2\varphi \right) \sin\varphi - \chi_c \sin 2\varphi \cos\varphi \right|^2 & P \perp A \end{cases} \quad (1)$$

where, $\chi_a = \chi^{m}_{xxy} + \chi^{e}_{xxx} + \Delta\chi^{e}_{xxx}$, $\chi_b = -\chi^{m}_{xyx} + \chi^{e}_{xyy} + \Delta\chi^{e}_{xyy}$, and $\chi_c = \left( \chi^{m}_{yyy} - \chi^{m}_{yxx} \right)/2 + \chi^{e}_{yxy} + \Delta\chi^{e}_{yxy}$. Note that Eq. (1) is common to the magnetic space group C2'/m at zero-field and Cm under a finite electric field. The ΔED contributions, $\Delta\chi^{e}_{xxx}(E)$, $\Delta\chi^{e}_{xyy}(E)$, and $\Delta\chi^{e}_{yxy}(E)$, are in first approximation proportional to the static electric field $E$, while the ED and MD components, $\chi^{e}_{ijk}$ and $\chi^{m}_{ijk}$, do not depend on $E$. By fitting our data with Eq. (1) under the constraint that $\chi_a$, $\chi_b$, and $\chi_c$ depend linearly on $E$ simultaneously, the complex $\chi_{a,b,c}(E)$ values are

determined with their phases referenced to the phase of $\chi_c(E=0)$. Additional details on the fitting analysis can be found in Supplementary Note 2. The resulting $\chi_{a,b,c}$ for Domains A and B are shown on the complex plane in Fig. 4a, b, respectively. Arrows indicate the linear paths of $\chi_{a,b,c}$ from negative to positive $E$, determined by $\Delta\chi^{e}_{ijk}$. Remarkably, all three $\chi_{a,b,c}$ follow linear trajectories on the complex plane, confirming the validity of this analysis. Moreover, the electric field is found to have a greater effect on the amplitude of $\chi_a$ compared to those of $\chi_b$ and $\chi_c$, in line with the more pronounced extension and shrinkage of the horizontal lobes in the SHG-RA of both Domains A and B for P ∥ A, than the vertical ones for P ⊥ A, in Fig. 2a.

The experimental results in Fig. 4a, b correspond to the introduced interferences in Fig. 3c, d. Specifically, for all three fitting coefficients $\chi_a$, $\chi_b$, and $\chi_c$, representing independent sets of multipolar source term interferences, the empty symbol (or approximately the midpoint of the arrow) at $E=0$ is located in opposite quadrants for Domains A and B in Fig. 4a, b, in line with the placement of Point 2 in Fig. 3c, d. Moreover, for each $\chi_{a,b,c}$, the arrows for Domains A and B in Fig. 4a, b, representing the trajectory of ΔED contributions, are nearly of the same length and direction, reproducing the situation depicted in Fig. 3c, d. Note that despite the potential difference in the reference phases for the two domains, the simultaneous satisfaction of opposite-sign zero-field $\chi$-values and similar/parallel trajectories for different domains described above demonstrates the agreement between experimental results and the proposed scheme. Thus, effective electrical control of domain contrast by biasing antiferromagnetic 180° domains in the opposite sense is fulfilled in this |ED| > |MD| regime. Additional results for the |MD| > |ED| regime at a $d$-$d$ transition resonance, with the fundamental light wavelength of 920 nm, can be found in Supplementary Note 3, where the analysis reveals an intensity modulation in the same sense for the two antiferromagnetic 180° domains, consistent with our anticipation of the process.

### Quantifying effects of electrical control
SHG domain contrast and nonreciprocity are further evaluated quantitatively at various electric fields and temperatures under the same $p$-$d$ charge-transfer resonance (Fig. 5). We define an SHG domain contrast ($\xi$) and nonreciprocity ($\eta$) as in Eqs. (2) and (3), applicable to both P ∥ A, $\varphi = 0°$ and P ⊥ A, $\varphi = 90°$ configurations.

**Table 1 | Space-inversion and time-reversal symmetries of the multipole $\chi_{ijk}$ tensors for the case of MnPS₃, with $L$ being the antiferromagnetic order parameter**

| Origin | Tensor | Type |
|---|---|---|
| MD | $\chi^{m}_{ijk} \left( \propto a_0 + a_2 L^2 \right)$ | axial, i |
| ED | $\chi^{e}_{ijk} (\propto L)$ | polar, c |
| ΔED | $\Delta\chi^{e}_{ijk} \left( = \chi^{e}_{ijkl} E_l \right)$ | polar, i |

$$\xi(E) = \frac{I^{A,E;k} - I^{B,E;k}}{I^{A,E;k} + I^{B,E;k}} \quad (2)$$

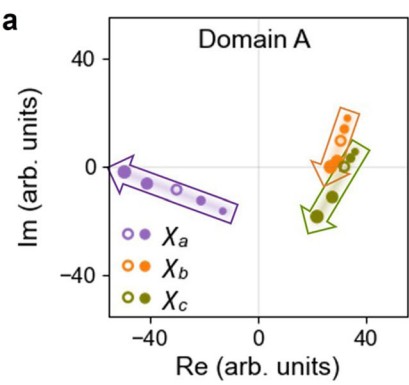

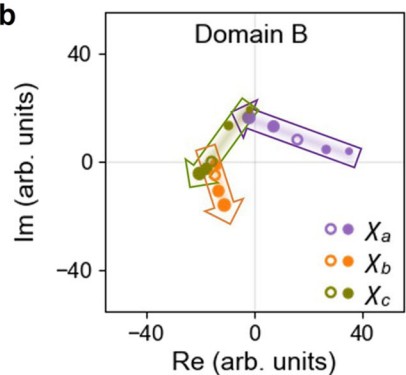

**Fig. 4 | Susceptibilities $\chi_{a,b,c}$, which are the results of multipolar source term interferences, displayed on the complex plane as a function of the applied electric field. a, b** Complex $\chi_{a,b,c}$ values for Domains A and B, referenced to the phase of $\chi_c$ at $E=0$. Colored dots, sized from small to large, represent $\chi_{a,b,c}$ at $E = -1.6, -0.8, 0, +0.8$, and $+1.6\,V/\mu m$, with shadings and arrows indicating their

evolution trends. Each $\chi_{a,b,c}$ corresponds to a set of interference in Fig. 3c, d, where Point 2 ($E=0$) in those figures corresponds to the empty dots here. The simultaneous fulfillment of zero-field $\chi$-values in opposite quadrants and similar trajectories for the two domains matches the scheme introduced in Fig. 3c, d.

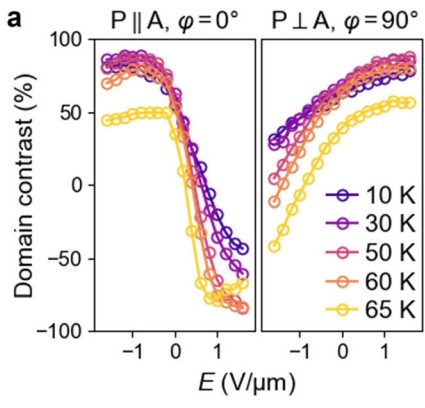
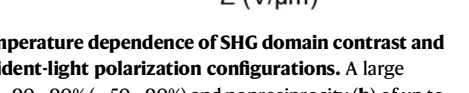
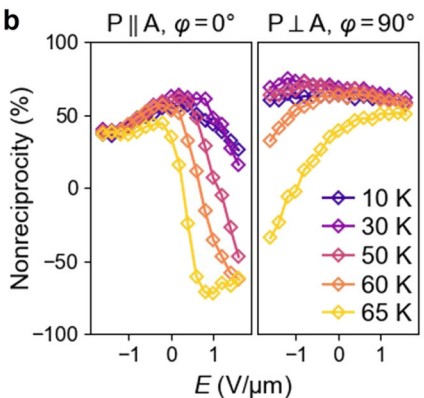

**Fig. 5 | Electric field and temperature dependence of SHG domain contrast and nonreciprocity for two incident-light polarization configurations.** A large domain contrast (**a**) of up to −90 ~ 90% (−50 ~ 90%) and nonreciprocity (**b**) of up to −70% ~ 70% (−40% ~ 70%) can be achieved in the P ∥ A, $\varphi = 0°$(P ⊥ A, $\varphi = 90°$) configuration. Reversal of domain contrast and nonreciprocity is achieved for both polarization configurations.

$$\eta(E) = \frac{I^{A,E;k} - I^{A,E;-k}}{I^{A,E;k} + I^{A,E;-k}} = \frac{I^{A,E;k} - I^{B,-E;k}}{I^{A,E;k} + I^{B,-E;k}} \qquad (3)$$

Here, $I$ represents the SH intensity under the [Domain, electric field; incident wavevector] condition noted in its superscript. The temperature dependence of $\xi(E)$ and $\eta(E)$ arises from the distinct relationships between the MD, ED, and ΔED $\chi$-tensors with the antiferromagnetic order parameter ($L$), which is a function of temperature. Specifically, the i-type $\chi_{ijk}^m\left(\propto a_0 + a_2 L^2\right)$ of MD SHG contains even powers of $L$ ($a_0$ and $a_2$ are temperature-dependent coefficients), while the c-type $\chi_{ijk}^e(\propto L)$ of ED SHG is proportional to $L$ to the lowest order[26]; these $L$-dependences are well-documented for $Cr_2O_3$ with broken space-inversion ($\mathcal{P}$) and time-reversal ($\mathcal{T}$) symmetries but preserved $\mathcal{PT}$-symmetry, the same as that of $MnPS_3$. The $\Delta\chi_{ijk}^e(\propto E)$ of the ΔED term should be mostly independent of $L$, being i-type, although a potential $E \cdot L$-dependent correction arising from the inherent linear magnetoelectric effect is allowed[26]. With combined electric-field and temperature effects, the domain contrast can be tuned in a wide range from −90% (−50%) to 90% (Fig. 5a), and considerable controllability of nonreciprocity is achieved from −70% (−40%) to 70% (Fig. 5b), for the two polarization configurations. We note that different from domain contrast, SHG nonreciprocity can appear even at temperatures above $T_N$ (~71 K for the thin flake) in the presence of the electric field. This occurs because of the interference between the residual crystallographic MD SHG, $\chi_{ijk}^m\left(\propto a_0 + a_2 L^2\right)$ at $L = 0$, and the ΔED term $\Delta\chi_{ijk}^e(\propto E)$, stemming from their distinct polar/axial characteristics. The nonvanishing crystallographic MD SHG in the paramagnetic phase has also been observed previously[14,21,22].

## Discussion

By directly imaging antiferromagnetic 180° domains in AFM $MnPS_3$ via SHG, we have demonstrated the effective enhancement and reversal of domain contrast, as well as the broad tunability of nonlinear optical nonreciprocity by applying electric fields. The proposed three-term interference scenarios can be naturally extended to other 2D and 3D systems with similar symmetries, potentially serving as a prototypical approach for the electrical control of nonreciprocity. Specifically, by applying an electric field that transforms the original magnetic point group into its unitary subgroup, the invariance of tensor forms for both ED and MD processes is maintained. This is exemplified by the shared SH intensity expression Eq. (1) for groups C2′/m ($E = 0$) and Cm ($E \neq 0$) in the present case; the ΔED term $\Delta\chi_{ijk}^e$, with the same $ijk$ indices as $\chi_{ijk}^e$, combines with the existing ED and MD terms in each $\chi_{a,b,c}$

coefficient, thereby ensuring the three-term interference. We suggest referring to such electrical control as the interaction-symmetry-preserved type, contrasting with interaction-symmetry-non-preserved ones where an electric field only activates new ED SHG tensor elements (see also Supplementary Note 4 for further exemplification of their distinctions). This interaction-symmetry-preserved electrical modulation can be broadly applied to systems in which the zero-field magnetic point group exhibits $\mathcal{PT}$-symmetry with its unitary subgroup being polar. More information, including a list of conforming magnetic point groups and the corresponding electric field directions, is provided in Supplementary Note 5.

Our strategy offers distinct advantages. Firstly, it operates independently of the need for a specific narrow MD transition, in contrast to many previous reports[27–30], thus generically providing broadband feasibility. Secondly, unlike the phase-sensitive interferometry technique for domain contrast enhancement that relies on external optics[31], our internal modulation approach directly alters the microscopic SH origins within the material itself. This methodology achieves significant domain contrast and nonreciprocity modulation through an electric field solely, overcoming limitations related to device size, operation speed, and polarization degree of freedom, imposed by the use of external phase-reference materials and phase-shifting optics. Thirdly, in comparison to the magnetic-field-based nonreciprocity control[27,28], electric-field control offers a much faster response and simpler device-architectural requirements. Also, given the inherent potential for miniaturization with 2D vdW materials, this work may illuminate the path toward active directional photon up-conversion in advanced photonic devices.

## Methods
### Sample and device preparation
$MnPS_3$ single crystals were prepared by a chemical vapor transport method[32], and exfoliated onto a 0.5-mm-thick sapphire (0001) substrate transparent in the visible and near-infrared spectral range. An optically uniform flake with a thickness of approximately 90 nm was chosen for measurements, with its topography shown in Supplementary Fig. 1. Au/Ti electrodes were deposited onto the flake and substrate using photolithography.

### SHG microscopy
The device was mounted in an optical cryostat with front and back windows (Janis ST-500) for transmission SHG measurements. A regenerative amplifier system that produces 190-fs laser pulses at a 6 kHz repetition rate was used as the light source. The fundamental light wavelength was tuned by an optical parametric amplifier (OPA). A water-cooled Electron Multiplying Charge Coupled Device (EM-CCD)

camera was employed for imaging the SH light with an objective lens (Olympus SLMPLN50X). The SHG images were acquired with an incidence pulse energy of less than $0.5\,\mu J$ (on a spot size of $120\,\mu m$ diameter).

## Data availability

All data used to generate the figures in the manuscript and Supplementary Information are available on Zenodo at: https://doi.org/10.5281/zenodo.13315907.

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

## Acknowledgements

Z.W. was supported by JSPS KAKENHI Grant No. 21K13889 and RIKEN Incentive Research Projects. Y.S. was supported by JSPS KAKENHI Grant No. 22H05449. N.N. and Y.T. were supported by CREST-JST (JPMJCR1874). N.O. was supported by JSPS KAKENHI Grant No. 22H01185. This work was supported by the RIKEN TRIP initiative (Multi-Electron Group). The authors declare no competing financial interests.

## Author contributions

Z.W. and N.O. conceived the project, performed the SHG measurements, and analyzed the data. M.W. performed the microfabrication and the atomic force microscopy measurements. J.L. contributed to the SHG measurements and data analysis. Y.S. grew the bulk single crystal. Z.W. wrote the manuscript, with contributions from J.L. and N.O. All authors collaborated in interpreting the data.

## Competing interests

The authors declare no competing interests.
