## [Peer Review File · Nature Communications]

Reviewers' Comments:

Reviewer #1:

Remarks to the Author:

Report on the manuscript 'Electric-field-enhanced second-harmonic domain contrast...'

by Ziqian Wang et al,*

Antiferromagnetic (AFM) sulfur phosphates of $3d^n$ transition metal ions, including Mn^{2+} , have become the subject of active research in the last decade due to potential applications in the field of AFM spintronics. The manuscript by Wang et al reports the experimental observation of nonlinear optical effect of second harmonic generation (SHG) in two-dimensional MnPS₃ with centrosymmetric monoclinic point symmetry $2/m$ (space symmetry $C2/m$). Below the Néel temperature $T_N = 78$ K (in a bulk state) the crystal is transformed into a honey-comb antiferromagnet, and the magnetic point group symmetry is $C2'm$. Experiment were performed on a sample in a form of nanometer thin flake of MnPS₃ grown on a sapphire substrate. The electric field E was applied in the (ab) sample plane.

The main result of the manuscript is an increase of the contrast between domains with the opposite orientation of the AFM order parameter L . There is a noticeable contrast in the sample between domains in zero applied electric field, but the application of the field leads to the contrast increase as it is demonstrated in Fig. 2. The authors estimate contrast increase up to $\pm 90\%$ using relations (3). This is accompanied by the nonreciprocity when changing the direction of the light propagation through the sample. For explaining the observed effects, the authors involve the interference between the three different sources of SHG, namely the intrinsic magnetic dipoles (MD) source which is allowed by symmetry both in the AFM and paramagnetic regions, and additionally, the two electric dipole sources (ED) and ΔED that appear due to an applied electric field in the AFM phase..

I have several questions about major claims in the manuscript.

1. It is claimed that AFM ordering in MnPS₃ breaks the crystallographic centrosymmetric monoclinic point group $2/m$, where the ED processes are forbidden, to the magnetic point group $2'm$ and therefore breaks the inversion symmetry: '...the antiferromagnetic ordering of Mn^{2+} ions ($S = 5/2$) breaks the inversion symmetry, resulting in magnetic space group $C2'm23...$ ' (page 3).

If I understand right, the AFM ordering breaks the time reversal symmetry but in zero electric field the crystallographic structure remains centrosymmetric. Is this statement about inversion breaking correct?

2. Breaking the centrosymmetric structure takes place when only an electric field is applied and the structure is transformed to an unitary subgroup. What is this subgroup?

3. How one can explain large contrast between domains in zero electric field $E=0$ when only MD source of SHG is allowed and no interference occurs? Can the A and B regions

in Fig. 2 be not AFM domains but crystallographic twins?

4. ED and MD usually differ in phase by 90 deg., but in Fig. 3 (c,d,e) they are shown with the same phase. Is this diagram correct?

5 Table I shows that the ED contribution to SHG is proportional to the AFM order parameter L which is expected to become larger at low temperature. However, Fig. 5 shows that the domain contrast is smaller at $T = 10$ K and larger at $T = 65$ K close to the transition into paramagnetic state where the L effects are small. How one can explain this observation?

6. What contributions to the observed effects of AFM domain contrast are expected from the electro-optical linear Pockels and quadratic Kerr effects? Can these sources interfere with the symmetry-allowed MD source, and therefore explain the observed interference effects of domain contrast and nonreciprocity?

In general, the manuscript is well written and well illustrated with detailed Figures. Some details are explained in Supplementary part.

It can be accepted for publication in Nature Communications after the authors give well-reasoned replies to the questions posed above.

Reviewer #2:

Remarks to the Author:

The interference between the c-type and i-type second-harmonic generation (SHG) has been demonstrated to enable the direct imaging of antiferromagnetic (AFM) domains (e.g., Ref. 22 for MnPS₃ and Ref. 2). In this work, Wang et al. propose that the electric field can enhance the SHG contrast of different domains through electric field-induced electric-dipole transitions. The experimental outcomes are trustworthy, and the writing is straightforward and comprehensible.

Regrettably, I cannot accept the current version of the manuscript for publication in Nature Communications. I have several questions regarding this work that I would like to discuss.

1. The authors state that this approach can be applied to a broad spectral range and to PT-symmetric systems with a magnetic point group featuring a polar unitary subgroup. First, I believe that any system should evolve into a polar group under the electric field. Second, why the polar unitary subgroup is so important to the electric field-induced electric-dipole transitions?

2. The authors claim that, under applied electric fields, the patterns maintain mirror symmetry along the horizontally aligned a -axis, which aligns with the expectations based on the magnetic space group C_m . What would be the outcome if the electric field

were applied along the b -axis? Since it should be a polar group under electric field, as mentioned in Comment 1.

3. Why are electric field-induced electric-dipole transitions observed instead of magnetic-dipole transitions?

4. The author should provide a more comprehensive discussion regarding the efficiency of the electric-field-enhanced SHG domain contrast methodology. What are the requirements for electric dipole (ED) and magnetic dipole (MD) moments in magnetic materials? Can the methodology be applied to MnPSe₃ and CrI₃, considering that the ED SHG is significantly larger than the MD SHG in these materials?

5. Why is the domain contrast of electrically induced SHG always better in Co- (P||A) polarization configurations compared to cross- (P⊥A) configurations, as demonstrated in Figure 2 and Supplementary Figure 4?

6. Why does the nonreciprocity of second-harmonic generation (SHG) persist even at temperatures above T_N in the presence of an electric field? According to the conventional concept, the two magnetic domain structures should be very similar above the Neel temperature, implying that the SHG signals should also be very similar.

We sincerely appreciate the thorough review and the generally approving comments, as well as the valuable suggestions that enhance the quality and clarity of our manuscript. Additionally, we are grateful for the opportunity to resubmit an improved version. We have highlighted all the changes in the revised manuscript and provided point-by-point replies to the reviewers' comments below. We believe these changes have significantly improved the manuscript's quality and are hopeful that it now meets all the requirements for publication in *Nature Communications*.

Reviewer #1 (Remarks to the Author):

Report on the manuscript 'Electric-field-enhanced second-harmonic domain contrast...' by Ziqian Wang et al,*

Antiferromagnetic (AFM) sulfur phosphates of $3d^n$ transition metal ions, including Mn^{2+} , have become the subject of active research in the last decade due to potential applications in the field of AFM spintronics. The manuscript by Wang et al reports the experimental observation of nonlinear optical effect of second harmonic generation (SHG) in two-dimensional $MnPS_3$ with centrosymmetric monoclinic point symmetry $2/m$ (space symmetry $C2/m$). Below the Néel temperature $T_N = 78$ K (in a bulk state) the crystal is transformed into a honey-comb antiferromagnet, and the magnetic point group symmetry is $C2'/m$. Experiment were performed on a sample in a form of nanometer thin flake of $MnPS_3$ grown on a sapphire substrate. The electric field E was applied in the (ab) sample plane.

The main result of the manuscript is an increase of the contrast between domains with the opposite orientation of the AFM order parameter L . There is a noticeable contrast in the sample between domains in zero applied electric field, but the application of the field leads to the contrast increase as it is demonstrated in Fig. 2. The authors estimate contrast increase up to $\pm 90\%$ using relations (3). This is accompanied by the nonreciprocity when changing the direction of the light propagation through the sample.

For explaining the observed effects, the authors involve the interference between the three different sources of SHG, namely the intrinsic magnetic dipoles (MD) source which is allowed by symmetry both in the AFM and paramagnetic regions, and additionally, the two electric dipole sources (ED) and ΔED that appear due to an applied electric field in the AFM phase.

Reply: We would like to extend our sincere gratitude to the reviewer for his/her meticulous review, comprehensive summary, and overall positive judgement on our work. Additionally, we greatly appreciate the valuable suggestions provided by the reviewer to enhance the quality and clarity of our manuscript. We have carefully considered all the feedbacks and have made revisions accordingly. We trust that our revised manuscript and responses provided below address all concerns raised by the reviewer.

I have several questions about major claims in the manuscript.

1. It is claimed that AFM ordering in MnPS_3 breaks the crystallographic centrosymmetric monoclinic point group $2/m$, where the ED processes are forbidden, to the magnetic point group $2'/m$ and therefore breaks the inversion symmetry:

‘...the antiferromagnetic ordering of Mn^{2+} ions ($S = 5/2$) breaks the inversion symmetry, resulting in magnetic space group $C 2'/m^{23} \dots$ ’ (page 3).

If I understand right, the AFM ordering breaks the time reversal symmetry but in zero electric field the crystallographic structure remains centrosymmetric. Is this statement about inversion breaking correct?

Reply: We confirm that the reviewer’s descriptions, “the AFM ordering breaks the time reversal symmetry” and “the crystallographic structure remains centrosymmetric (in the AFM phase),” align with our understanding. However, we would like to point out that the conjunction of these facts does not inherently conclude the (space-)inversion breaking in our original statement, as the isostructural compound FePS_3 is an example that antiferromagnetic ordering does not break space-inversion symmetry (Ref. R1). Allow us to elaborate on the space-inversion breaking in MnPS_3 further.

As generally defined, *crystal structure* refers to the arrangement of atoms without considering spins, while *magnetic structure* refers to the arrangement of spins on the crystal lattice. The material is a combination of its crystal structure and magnetic structure. In the antiferromagnetic phase of MnPS_3 at zero electric field, although the crystal structure has space-inversion symmetry (being centrosymmetric), the spin arrangement or magnetic structure does not coincide with itself under space-inversion operation (Fig. R1). Note that the space-inversion operation does not alter the spin direction for each atom, as spins are axial vectors. Thus, space-inversion symmetry is broken in this material due to magnetic ordering.

Fig. R1. Illustration of the space-inversion operation on antiferromagnetic MnPS_3 . Here, only Mn ions are shown for clarity. Such an operation cannot recover it to the original state due to the existence of magnetic order, indicating an inversion symmetry breaking. Such two states correspond to the antiferromagnetic Domain A and Domain B.

Such a situation, broken inversion symmetry due to magnetic ordering, has been known in many sample systems, including Cr_2O_3 (Ref. R2) and see, for example, “Nonlinear Optics on Ferroic Materials” by M. Fiebig (Wiley, 2024).

[R1] H. Chu, *et al.* Linear Magnetolectric Phase in Ultrathin MnPS_3 Probed by Optical Second

Harmonic Generation. Phys. Rev. Lett. 124, 027601 (2020)

[R2] M. Fiebig *et al.*, Second Harmonic Generation and Magnetic-Dipole-Electric-Dipole Interference in Antiferromagnetic Cr₂O₃. Phys., Rev. Lett. 73, 2127 (1994)

2. Breaking the centrosymmetric structure takes place when only an electric field is applied and the structure is transformed to an unitary subgroup. What is this subgroup?

Reply: For MnPS₃, this subgroup is the point group m . Point group m is the unitary subgroup of MnPS₃'s magnetic point group $2'/m$.

A type-III (or black-and-white) magnetic group M has the structure of $M = G + AG$. Here, G is the unitary subgroup of M , comprising solely the usual three-dimensional symmetry operations, while AG is its coset, containing only antiunitary operations involving time-reversal, or equivalently, spin-inversion (Refs. R3, R4). M can refer to a magnetic point group or a magnetic space group in the statement above. For the discussion of SHG, it is sufficient to use magnetic point groups, and Supplementary Table 1 presents such magnetic point group-unitary subgroup pairs that can be transformed by applying an electric field.

We have added the sentences “*The electric field transforms the magnetic space group from $C2'/m$ to Cm .*” to page 3 of the main text and “*Antiferromagnet MnPS₃ in this study corresponds to the parent group $2'/m$.*” to the Supplementary Note 5 for better clarity.

[R3] A. Cracknell, Crystal field theory and the Shubnikov point groups. Advances in Physics 17, 367-420 (1968)

[R4] C. Bradley and B. Davies, Magnetic Groups and Their Corepresentations. Rev. Mod. Phys. 40, 359-379 (1968)

3. How one can explain large contrast between domains in zero electric field $E=0$ when only MD source of SHG is allowed and no interference occurs? Can the A and B regions in Fig. 2 be not AFM domains but crystallographic twins?

Reply: At $E = 0$ below T_N , both ED SHG and MD SHG are symmetry-allowed, as described in the lines 10-12 in page 3, “*In this setting, both the ED process, $P_i(2\omega) \propto \chi_{ijk}^e E_j(\omega) E_k(\omega)$, and the MD process, $P_i(2\omega) \propto \chi_{ijk}^m E_j(\omega) H_k(\omega)$, contribute to the SHG response.*”, and the domain contrast arises from their interference. The observation of zero-field domain contrast is consistent with previous reports (Ref. 22).

In fact, a relocation of the antiferromagnetic domain walls occurs each time the temperature crosses T_N . We have included a new Supplementary Fig. 2 (replicated as Fig. R2 below), displaying images obtained from measurements after two separate cooling cycles in (a) and (b), respectively. Crystallographic twins typically have significantly less mobility at such low temperatures. Therefore, we conclude that the contrast arises from antiferromagnetic domains rather than structural defects such as crystallographic twins.

We have also modified our descriptions on page 3 of the main text accordingly: “*Regions A and B, corresponding to higher and lower SH intensities in the absence of electric field, are*

identified as antiferromagnetic 180° domains, as confirmed by their emergence below T_N with varying shapes and locations after each cooling cycle (see Supplementary Fig. 2 for examples).”

Fig. R2 (new Supplementary Fig. 2). Variation in domain morphology after different cooling cycles. a,b SHG images acquired after separate cooling events through T_N , at a sample temperature of 10 K using a fundamental wavelength of 840 nm. Domain walls are indicated by white dashed lines for clarity. Images in (a) correspond to those in Fig. 2. This observation indicates that the contrast arises from antiferromagnetic 180° domains rather than structural defects like crystallographic twins, which are much less mobile.

4. ED and MD usually differ in phase by 90° , but in Fig. 3 (c,d,e) they are shown with the same phase. Is this diagram correct?

Reply: We agree with the reviewer that ED and MD SHG usually have phase difference of 90° in *off-resonance conditions*. However, this phase difference is not necessarily satisfied in the present case with a *resonant SHG processes*. Also, we would like to remind the reviewer that the diagrams are of conceptual nature only as mentioned in the figure caption and the end of page 4 in the main text.

Equations R1 and R2 presents the general microscopic expression for ED and MD SHG susceptibilities (Ref. R5, R6):

$$\chi_{ijk}^{eee} \propto \sum_{g,n,n'} \left[\frac{\langle g|P_i|n\rangle \langle n|P_j|n'\rangle \langle n'|P_k|g\rangle}{(2\omega - \omega_{ng} + i\Gamma_{ng})(\omega - \omega_{n'g} + i\Gamma_{n'g})} + \text{other 7 terms} \right] \quad (\text{R1})$$

$$\chi_{ijk}^{eem} \propto \sum_{g,n,n'} \left[\frac{\langle g|P_i|n\rangle \langle n|P_j|n'\rangle \langle n'|M_k|g\rangle}{(2\omega - \omega_{ng} + i\Gamma_{ng})(\omega - \omega_{n'g} + i\Gamma_{n'g})} + \text{other 7 terms} \right] \quad (\text{R2})$$

Here, g , n , n' represent ground and excited states involved in the resonant SHG process, Γ 's are damping constants, and $\langle g|P_i|n \rangle$ and similar terms represent transition dipole moments. When the fundamental and/or SH photon energy is in close vicinity of electronic transitions of the material, the complex denominators become sensitive to the damping constants Γ 's associated with each transition among g , n , n' states through ED or MD dipole transitions. This results in nontrivial phase relations between ED and MD SHG.

Due to the generally not-fully resolvable phase relations of ED, MD, and Δ ED terms, we schematically display their χ 's on a line in the complex plane for better illustration of various interfering scenarios, which, we believe, does not change the physics we discussed.

[R5] Y. R. Shen, *The Principles of Nonlinear Optics* (Wiley, New York, 1984).

[R6] M. Fiebig, et al. *Second Harmonic Generation in the Centrosymmetric Antiferromagnet NiO*. *Phys. Rev. Lett.* 87, 137202 (2001).

5. Table I shows that the ED contribution to SHG is proportional to the AFM order parameter L which is expected to become larger at low temperature. However, Fig. 5 shows that the domain contrast is smaller at $T = 10$ K and larger at $T = 65$ K close to the transition into paramagnetic state where the L effects are small. How one can explain this observation?

Reply: We appreciate the reviewer's scrutiny of our experimental results. However, we would like to point out that in Fig. 5a, at $E = 0$, the domain contrast at 10 K is actually higher than at 65 K, contrary to the reviewer's description, for both $P \parallel A, \varphi = 0^\circ$ and $P \perp A, \varphi = 90^\circ$ configurations. For clarity, we have replotted the domain contrast at $E = 0$ as functions of temperature in Fig. R3a.

The temperature dependence of domain contrast arises from the distinct temperature-dependent behaviors of the ED [$\chi_{ijk}^e(\propto L)$] and MD [$\chi_{ijk}^m(\propto a_0 + a_2 L^2)$] SHG contributions (Table 1). This dependence is clarified by simple linear and quadratic function plots, as depicted in Fig. R3b. Specifically:

- (0) Above T_N (~ 70 K), the domain structure is absent and therefore domain contrast is undefined.
- (1) Just below T_N , where the antiferromagnetic order parameter L is close to 0, the domain contrast is near zero due to the negligible χ_{ijk}^e compared to χ_{ijk}^m .
- (2) With a slight decrease in temperature, such as at 65 K, where L has a small value and both χ_{ijk}^e and χ_{ijk}^m are finite, the domain contrast becomes non-zero.
- (3) Further temperature decrease brings χ_{ijk}^e closer to χ_{ijk}^m , resulting in a slight increase in the domain contrast, i.e., contrast between $|\chi_{ijk}^e + \chi_{ijk}^m|^2$ and $|\chi_{ijk}^e - \chi_{ijk}^m|^2$, as long as their phase difference is not perfectly 90° .
- (4) At sufficiently lower temperatures, like 10 K, the significant L^2 -dependence of χ_{ijk}^m leads to a larger separation between χ_{ijk}^e and χ_{ijk}^m , resulting in a smaller domain contrast.

We have increased the spacing between the two panels in Figs. 5a and 5b in the main text to improve clarity.

Fig. R3. Illustration of the temperature dependence of SHG domain contrast. (a) Domain contrast plotted against temperature at $E = 0$. Both $P \parallel A, \varphi = 0^\circ$ and $P \perp A, \varphi = 90^\circ$ configurations exhibit a trend of an increase followed by a decrease in domain contrast as temperature decreases (L increases). (b) Linear and quadratic functions representing the temperature dependence of χ_{ijk}^e and χ_{ijk}^m . Status (2) (3) (4) are noted in the figures.

6. What contributions to the observed effects of AFM domain contrast are expected from the electro-optical linear Pockels and quadratic Kerr effects? Can these sources interfere with the symmetry-allowed MD source, and therefore explain the observed interference effects of domain contrast and nonreciprocity?

Reply: We appreciate the reviewer's thoughtful comment. The electro-optical linear Pockels and quadratic Kerr effect describe changes in the refractive index of a material, or induced optical uniaxiality, with linear and quadratic dependences on the external electric field, respectively. Therefore, these effects do not generate additional light-emitting sources to directly interfere with SHG sources.

While it is conceivable that an electric field could make slight orientation-dependent modifications to the refractive indices at ω and 2ω through these effects, such changes will, in principle, be consistent across different antiferromagnetic domain regions. Therefore, any potential minor influences on the absolute SH intensity from these effects should not affect the SH domain contrast and nonreciprocity defined based on the intensity ratios. Moreover, the slight changes in phase matching conditions for SHG (or effects from so-called coherence length), due to potential refractive index changes from these effects, are considered to be negligible in our transmission geometry, since the sample thickness is well below the light wavelength involved. Thus, we consider that these two effects cannot explain our observations.

In general, the manuscript is well written and well illustrated with detailed Figures. Some details are explained in Supplementary part.

It can be accepted for publication in Nature Communications after the authors give well-reasoned replies to the questions posed above.

Reply: We express again our gratitude to the reviewer once more for his/her diligent review and positive evaluation of our work. We hope that our explanations above have addressed all the

reviewer's concerns.

Reviewer #2 (Remarks to the Author):

The interference between the c-type and i-type second-harmonic generation (SHG) has been demonstrated to enable the direct imaging of antiferromagnetic (AFM) domains (e.g., Ref. 22 for MnPS₃ and Ref. 2). In this work, Wang et al. propose that the electric field can enhance the SHG contrast of different domains through electric field-induced electric-dipole transitions. **The experimental outcomes are trustworthy, and the writing is straightforward and comprehensible.**

Regrettably, I cannot accept the current version of the manuscript for publication in Nature Communications. I have several questions regarding this work that I would like to discuss.

Reply: We would like to convey our sincere appreciation to the reviewer for his/her essential summary of our work and the positive evaluation of our results. Additionally, we are grateful for the valuable suggestions and comments provided to enhance the quality of our manuscript and for offering a chance for resubmission. We believe that all concerns raised by the reviewer have been thoroughly addressed in our revised manuscript and responses provided below.

1. The authors state that this approach can be applied to a broad spectral range and to PT-symmetric systems with a magnetic point group featuring a polar unitary subgroup. First, I believe that any system should evolve into a polar group under the electric field. Second, why the polar unitary subgroup is so important to the electric field-induced electric-dipole transitions?

Reply: Regarding the first point, we agree with the reviewer that applying an electric field to a material naturally transforms the corresponding magnetic point group into one of its polar subgroups. However, it is important to note that while this polar subgroup generically allows for electric-field-induced ED (Δ ED) SHG, the Δ ED component may not interfere “effectively” with the existing ED and MD SHG components. In our proposal, the “effective” condition (page 7, main text) which allows for maximally favored ED-MD- Δ ED interference is, from the group-theoretical viewpoint, the *consistency of symmetry selection rules for SHG tensor elements before and after electric field application*. This condition is met when the polar subgroup coincides with the unitary subgroup of the original magnetic point group.

Specifically, for antiferromagnetic MnPS₃, an electric field along the *a*-axis transforms the magnetic point group $2'/m$ to its unitary subgroup m . In this configuration, the Δ ED contribution arises with tensor $\Delta\chi_{ijk}^e$ (active elements: $\Delta\chi_{xxx}^e$, $\Delta\chi_{xyy}^e$, $\Delta\chi_{yxy}^e = \Delta\chi_{yyx}^e$), which shares the same indices as χ_{ijk}^e (active elements: χ_{xxx}^e , χ_{xyy}^e , $\chi_{yxy}^e = \chi_{yyx}^e$) of the ED component at zero field. Consequently, Δ ED SHG emerges *along the symmetry-allowed directions of zero-field SHG*, as depicted in Supplementary Fig. 7a. The Δ ED SHG, indicated schematically by the red dashed curves, interferes with zero-field SHG (left panels), leading to the observed finite-field SHG-RA patterns (right panels). Since the maxima of Δ ED and zero-field SHG are aligned, the ED-MD- Δ ED

interference is maximally “effective” from the symmetry viewpoint.

In contrast, for an electric field applied along the b -axis, the magnetic point group is transformed to $2'$, a polar subgroup distinct from the unitary one. In this scenario, the Δ ED tensor elements ($\Delta\chi_{yxx}^e, \Delta\chi_{yyx}^e, \Delta\chi_{xxy}^e = \Delta\chi_{xyx}^e$) differ in indices from the ED ones ($\chi_{xxx}^e, \chi_{xyy}^e, \chi_{yxy}^e = \chi_{yyx}^e$), resulting in Δ ED SHG aligning with *symmetry-forbidden directions of SHG*, and thus suppressed, as illustrated in Supplementary Fig. 7b. Specifically, although extension and contraction of lobes are noticeable at polarization angles of 60, 120, 240, and 300°, the red dashed lobes of Δ ED point to directions where SH intensity is zero, suggesting that the ED-MD- Δ ED interference is primarily suppressed. In this sense, the condition is referred to as “ineffective” from the symmetry viewpoint.

Thus, the comparative examples provided above illustrate the importance of polar unitary subgroup in our framework. We have included this explanation in the Supplementary information as the new Supplementary Note 4.

Fig. R4 (new Supplementary Fig. 7). Comparison of interference scenarios under electric fields in different directions. a SHG-RA patterns before (left) and after (right) applying an electric field along the a -axis. These patterns correspond to those for Domain A at 10 K in Fig. 2a in the main text. **b** SHG-RA patterns for electric field along the b -axis, obtained from a different specimen. The red dashed curves schematically represent the pure electric-field-induced contributions, which interfere with zero-field SHG (left) and produce the finite-field SHG-RA patterns (right).

2. The authors claim that, under applied electric fields, the patterns maintain mirror symmetry along the horizontally aligned a -axis, which aligns with the expectations based on the magnetic space group C_m . What would be the outcome if the electric field were applied along the b -axis? Since it should be a polar group under electric field, as mentioned in Comment 1.

Reply: We have elaborated on this point in conjunction with our response to Comment #1.

3. Why are electric field-induced electric-dipole transitions observed instead of magnetic-dipole

transitions?

Reply: MD SHG tensors χ_{ijk}^m are *axial* tensors and thus do not couple to the electric field E (*polar* vector) with an odd exponent. Therefore, the lowest possible order of E in χ_{ijk}^m would be E^2 . However, such high-order terms are typically of very small amplitude and are thus considered negligible.

4. The author should provide a more comprehensive discussion regarding the efficiency of the electric-field-enhanced SHG domain contrast methodology. What are the requirements for electric dipole (ED) and magnetic dipole (MD) moments in magnetic materials? Can the methodology be applied to MnPSe₃ and CrI₃, considering that the ED SHG is significantly larger than the MD SHG in these materials?

Reply: We appreciate the reviewer's engaging comments and questions. The efficiency of electric-field-controlled SHG domain contrast/nonreciprocity depends not only on the magnitudes of $\Delta\chi_{ijk}^e$ elements in response to the electric field but also on their phases relative to those of χ_{ijk}^e and χ_{ijk}^m . Both aspects bear a significant and intricate reliance on the resonance paths, which are inherently complicated due to the involvement of numerous excited states. Therefore, we find a generalization and a comprehensive discussion of the efficiency to be highly challenging.

Moreover, the electric and magnetic transition dipole moments involved in a certain SHG process, the $\langle g|\hat{\mathbf{P}}|n\rangle$ and $\langle n'|\hat{\mathbf{M}}|g\rangle$ etc. in Eqs. R1 and R2 (Ref. R5, R6), do not have a simple relationship with the ED and MD dipole moments, $\sum_n \langle n|\hat{\mathbf{P}}|n\rangle$ and $\sum_n \langle n|\hat{\mathbf{M}}|n\rangle$, of the material.

$$\chi_{ijk}^{eee} \propto \sum_{g,n,n'} \left[\frac{\langle g|P_i|n\rangle \langle n|P_j|n'\rangle \langle n'|P_k|g\rangle}{(2\omega - \omega_{ng} + i\Gamma_{ng})(\omega - \omega_{n'g} + i\Gamma_{n'g})} + \text{other 7 terms} \right] \quad (\text{R1})$$

$$\chi_{ijk}^{eem} \propto \sum_{g,n,n'} \left[\frac{\langle g|P_i|n\rangle \langle n|P_j|n'\rangle \langle n'|M_k|g\rangle}{(2\omega - \omega_{ng} + i\Gamma_{ng})(\omega - \omega_{n'g} + i\Gamma_{n'g})} + \text{other 7 terms} \right] \quad (\text{R2})$$

This complexity makes it challenging to define a requirement for the ED and MD moments in the material without detailed knowledge of its electronic structure. Therefore, we believe that the symmetry requirements presented in this manuscript serve as a more practical approach.

From a symmetry viewpoint, our methodology is applicable to antiferromagnetic MnPSe₃ and CrI₃, which have magnetic point groups $\bar{1}'$ and $2/m'$, featured by polar unitary subgroups 1 and 2, respectively (Supplementary Table 1). (Monolayer MnPSe₃ shares the same magnetic point group as MnPS₃.) Determining whether significantly greater ED SHG than MD SHG ensures high efficiency of electric field control is not straightforward, as the relative phases of $\Delta\chi_{ijk}^e$, χ_{ijk}^e , and χ_{ijk}^m elements depend on resonance conditions, i.e., material electronic structure and light wavelength. Nevertheless, the phases and amplitudes of the multipolar SHG sources, along with domain contrast and nonreciprocity, can be adjusted to some extent by varying the fundamental wavelength and temperature, as demonstrated in this work.

[R5] Y. R. Shen, *The Principles of Nonlinear Optics* (Wiley, New York, 1984).

[R6] M. Fiebig, et al. *Second Harmonic Generation in the Centrosymmetric Antiferromagnet NiO*.

Phys. Rev. Lett. 87, 137202 (2001).

5. Why is the domain contrast of electrically induced SHG always better in Co- ($P \parallel A$) polarization configurations compared to cross- ($P \perp A$) configurations, as demonstrated in Figure 2 and Supplementary Figure 4?

Reply: We appreciate the reviewer's thorough examination of our results and the insightful question posed. The different “efficiency” of the electric field effect in co- and cross- configurations primarily stems from the different behaviors of χ_a ($= \chi_{xxx}^e + \chi_{xxy}^m + \Delta\chi_{xxx}^e$), contributing to $P \parallel A, \varphi = 0^\circ$ (horizontal lobes in Fig. 2(a)), and χ_b ($= \chi_{xyy}^e - \chi_{xyx}^m + \Delta\chi_{xyy}^e$), contributing to $P \perp A, \varphi = 90^\circ$ (vertical lobes in Fig. 2(b)). As depicted in Fig. 4, the contribution of $\Delta\chi_{ijk}^e$ (hollow arrow) aligns more closely with the direction from the origin to the initial value (empty dot) in χ_a compared to χ_b . This results in a more efficient modulation of the χ_a 's amplitude than χ_b 's by the electric field. As elucidated in our response to Comment #4, this condition arises from the phase relationship among χ_{ijk}^e , χ_{ijk}^m , and $\Delta\chi_{ijk}^e$, which depends on the detailed electronic structure.

We have revised lines 17-20 on page 5 of the main text as follows to improve clarity: “Besides, the electric field is found to have greater effect on the amplitude of χ_a compared to those of χ_b and χ_c , in line with the **more pronounced extension and shrinkage of the horizontal lobes in the SHG-RA of both Domains A and B for $P \parallel A$, than the vertical ones for $P \perp A$, in Fig. 2a.**”

6. Why does the nonreciprocity of second-harmonic generation (SHG) persist even at temperatures above T_N in the presence of an electric field? According to the conventional concept, the two magnetic domain structures should be very similar above the Neel temperature, implying that the SHG signals should also be very similar.

Reply: Firstly, domain structure is absent above T_N . Even if no magnetic order remains, crystallographic MD SHG, i.e., the a_0 part of χ_{ijk}^m ($\propto a_0 + a_2 L^2$) in Table 1, can still exist, which may interfere with the electric-field-induced ED SHG (ΔED). Such crystallographic SHG has been known in many systems (Ref. R7). In this scenario, the forward and backward propagation of light leads to different SH intensities in the presence of a finite electric field, resulting in nonreciprocity. Reversing the direction of the electric field changes the sign of nonreciprocity. However, the remaining MD SHG is typically very weak above T_N in most cases.

[R7] M. Fiebig and R. V. Pisarev, Nonlinear optical spectroscopy and spin effects in magnetic compounds. Physica Status Solidi C: Conferences 1452, 1449-1452 (2003).

Again, we are sincerely grateful for the opportunity to enhance our manuscript based on the reviewer's valuable feedback. We hope that our revisions and responses above meet the reviewer's expectations and address all concerns of the reviewer.

Reviewers' Comments:

Reviewer #1:

Remarks to the Author:

The authors of the manuscript provided detailed, reasoned answers to all the questions in my review. I recommend accepting the manuscript 'Electric-field-enhanced second-harmonic domain contrast and nonreciprocity in a van der Waals antiferromagnet' for publication in Nature Communications.

Reviewer #2:

Remarks to the Author:

Thank you very much to the authors for providing direct answers to our questions, addressing most of our concerns. However, we still believe the polar unitary subgroup is not a critical factor for the electric field-induced electric-dipole transitions (i.e., Comment 1 and 2). An electric field applied along the b-axis can also induce a significant change in electric-dipole second-harmonic generation (Δ ED SHG), even though the Δ ED SHG indices differ from the electric-dipole (ED) ones. This Δ ED SHG can still interfere with zero-field second-harmonic generation, as shown in Supplementary Fig. 7. In this case, the second-harmonic generation-rotational anisotropy (SHG-RA) patterns do not clearly contrast in different antiferromagnetic (AFM) domains along the a-axis. However, I think this effect is very significant when considering polarization angles near 60° , 120° , 240° , and 300° . Therefore, the electric field applied along the b-axis should also be considered an "effective" electrical control direction.

We sincerely appreciate the second-round review and the valuable comments for enhancing the quality and clarity of our manuscript. Again, we are grateful for the opportunity to resubmit an improved version.

Reviewer #1 (Remarks to the Author):

The authors of the manuscript provided detailed, reasoned answers to all the questions in my review. I recommend accepting the manuscript 'Electric-field-enhanced second-harmonic domain contrast and nonreciprocity in a van der Waals antiferromagnet' for publication in Nature Communications.

Reply: We would again like to express our gratitude for the critical review and positive evaluation of our manuscript, and finally, the recommendation for acceptance.

Reviewer #2 (Remarks to the Author):

Thank you very much to the authors for providing direct answers to our questions, addressing most of our concerns. However, we still believe the polar unitary subgroup is not a critical factor for the electric field-induced electric-dipole transitions (i.e., Commet 1 and 2). An electric field applied along the b-axis can also induce a significant change in electric-dipole second-harmonic generation (Δ ED SHG), even though the Δ ED SHG indices differ from the electric-dipole (ED) ones. This Δ ED SHG can still interfere with zero-field second-harmonic generation, as shown in Supplementary Fig. 7. In this case, the second-harmonic generation-rotational anisotropy (SHG-RA) patterns do not clearly contrast in different antiferromagnetic (AFM) domains along the a-axis. However, I think this effect is very significant when considering polarization angles near 60° , 120° , 240° , and 300° . Therefore, the electric field applied along the b-axis should also be considered an "effective" electrical control direction.

Reply: We sincerely appreciate the reviewer's thorough assessment and insightful comments regarding the polar unitary subgroup and the "effective" electrical control direction. We agree that, in the case of MnPS_3 , the electric field along the *b*-axis also significantly modulates SHG near 60° , 120° , 240° , and 300° . However, this observation does not diminish the crucial role of polar unitary subgroups in establishing a *general strategy* for electrical control of SHG domain contrast and nonreciprocity based on symmetry. This is because whether the major direction of induced Δ ED SHG aligns with the forbidden direction of zero-field SHG is an important issue, which can be judged using the polar unitary subgroups. We would like to elaborate on this point by highlighting the following aspects.

First, modulating the lobes near 60° , 120° , 240° , and 300° (for $P \parallel A$) is not representative, as these lobes are not inherently expected from the $C 2'/m$ group. The lobes primarily stem from the approximate 3-fold rotation symmetry of each MnPS_3 layer. Figure R1 illustrates exemplary plots of the symmetry-adapted SH intensity formula for $C 2'/m$ (Eq. 1 in the main text) using different sets of numerical values for χ_a, χ_b, χ_c . When $(\chi_a, \chi_b, \chi_c) = (-1, 1, 1)$, the $P \parallel A$ SHG-RA pattern exhibits clear lobes at 0° , 60° , 120° , 180° , 240° , and 300° (Fig. R1a), similar to our observations for MnPS_3 . However, when $(\chi_a, \chi_b, \chi_c) = (1, -1, 1)$, the $P \parallel A$ SHG-RA pattern only shows two lobes at 0° and 180° (Fig. R1b). Thus, the lobes near 60° , 120° , 240° , and 300° are not guaranteed by the $C 2'/m$ symmetry, and such features cannot always serve as good targets for electrical control. Therefore, modulating features at these and other specific angles with an electric field different from the one in the "effective" case is considered a case-by-case scenario due to their lack of generality. This situation extends beyond the general strategy we intend to propose in this manuscript.

Fig. R1. Plots of symmetry-adapted SH intensity for the $C_{2'}/m$ group. (a) The case similar to our observations for MnPS_3 at zero field. (b) Another case showing no lobes in the 60°, 120°, 240°, and 300° directions. Lobes at 60°, 120°, 240°, and 300° in (a) are not inherently expected from the $C_{2'}/m$ group.

Second, our use of the terms “effective” and “ineffective” was in fact *not intend to evaluate effectiveness phenomenologically but rather to conceptually categorize scenarios of ED-MD- Δ ED interference*. To avoid confusion, we have replaced them with “interaction-symmetry-preserved” and “interaction-symmetry-non-preserved” in our revised manuscript, highlighting the symmetry aspect of SHG interactions (see paragraph “Discussion” on page 7 in the updated main text). In the “interaction-symmetry-non-preserved” or original “ineffective” scenario, the primary direction of Δ ED SHG aligns with symmetry-forbidden directions for zero-field SHG (Fig. R2b, 90 and 270° for $P \parallel A$). *The absence of intensity in the zero-field SHG in these directions makes it in general unsuitable for electrical control of domain contrast and nonreciprocity* (See page 9-10 in the updated Supplementary Information).

The key argument for our stated preference of the polar unitary subgroup lies in the straightforward distinction between “interaction-symmetry-preserved” and “interaction-symmetry-non-preserved” cases, from which only the former offers a general strategy with a predictable electric-field-controlled SHG response. Note, however, that *we refrain from claiming that the “interaction-symmetry-preserving” polar unitary subgroup constitutes the only pathway for an efficient electrical control of the SHG yield*.

Once again, we sincerely appreciate the reviewer’s valuable comments, which have significantly enhanced our manuscript. We hope that our revisions and responses above address all the concerns of the reviewer.

Fig. R2 (new Supplementary Fig. 7). Comparison of interference scenarios under electric fields applied along two different crystal axes. **a** SHG-RA patterns with (right) and without (left) application of an electric field along the a -axis, exemplifying an “interaction-symmetry-preserved” scenario. These patterns correspond to those for Domain A at 10 K in Fig. 2a in the main text. **b** SHG-RA patterns for electric field application along the b -axis, obtained from a different specimen, exemplifying an “interaction-symmetry-non-preserved” scenario. The red curves in (a) and (b) schematically represent the pure electric-field-induced contributions, which interfere with zero-field SHG and produce the finite-field SHG-RA patterns. **c** SHG tensor forms for magnetic groups $C2'/m$, Cm , and $C2'$ under $E = 0$, $E \parallel a$, and $E \parallel b$, respectively. Upper and lower represent χ_{ijk}^e for ED SHG and χ_{ijk}^m for MD SHG, respectively. Elements modulated or activated by an electric field are highlighted in red in the tensors for Cm or $C2'$.

Reviewers' Comments:

Reviewer #2:

Remarks to the Author:

I have reviewed the authors' response and believe that they have successfully addressed the weaknesses identified in the initial review. I am pleased to endorse its publication in Nat. Commun.

Reviewer #2 (Remarks to the Author):

I have reviewed the authors' response and believe that they have successfully addressed the weaknesses identified in the initial review. I am pleased to endorse its publication in Nat. Commun.

Reply: We would like to express again our gratitude for the thorough review, valuable comments, and the recommendation for acceptance.